# Unraveling the nature of physicochemical and biological processes underlying vesicular exocytotic release events through modeling of amperometric current spikes

## Perspective

amperometric spikes; fusion pore dynamics; organization of intravesicular matrixes; physicochemical modeling; single-cell amperometry; vesicular exocytosis

**Corresponding author:**
Alexander Oleinick;
Email: oleksandr.oliynyk@ens.psl.eu

## Alexander Oleinick[1] , Irina Svir[1] and Christian Amatore[1,2]

[1]Chimie Physique et Chimie du Vivant, Département de Chimie, Ecole Normale Supérieure, PSL Université, Sorbonne Université, CNRS, Paris, France and [2]State Key Laboratory of Physical Chemistry of Solid Surfaces, College of Chemistry and Chemical Engineering, Xiamen University, Xiamen, People's Republic of China

## Abstract

This work offers a comprehensive approach to understanding the phenomena underlying vesicular exocytosis, a process involved in vital functions of living organisms such as neuronal and neuroendocrine signaling. The kinetics of release of most neuromediators that modulate these functions in various ways can be efficiently monitored using single-cell amperometry (SCA). Indeed, SCA at ultramicro- or nanoelectrodes provides the necessary temporal, flux, and nanoscale resolution to accurately report on the shape and intensity of single exocytotic spikes. Rather than characterizing amperometric spikes using standard descriptive parameters (e.g., amplitude and half-width), however, this study summarizes a modeling approach based on the underlying biology and physical chemistry of single exocytotic events. This approach provides deeper insights into intravesicular phenomena that control vesicular release dynamics. The ensuing model's intrinsic parsimony makes it computationally efficient and friendly, enabling the processing of large amperometric traces to gain statistically significant insights.

## Introduction

Communication between cells and their environment is facilitated by several mechanisms, one of which is vesicular exocytosis, which serves vital functions in various systems (e.g., nervous, digestive, immune) in both humans and animals (see e.g., (Kandel et al., 2021)) as well as in plants (Liu et al., 2014). The characterization of individual vesicular exocytotic release events is challenging due to the minute dimensions of vesicles (ranging from tens to hundreds of nanometers, depending on the cell type) and their short duration (from milliseconds to a few hundred milliseconds). Single-cell amperometry (SCA), with its high temporal resolution and the ability to quantify extremely small numbers (from tens of thousands to millions) of released molecules by using Faradays law, is a well-established technique for characterizing individual exocytotic events in different cell types (Figure 1) (Schroeder et al., 1992; Amatore et al., 2008). Amperometric traces recorded from single living cells appear as sequences of asymmetric current spikes, featuring individual exocytotic release events corresponding to the release of their cargo of molecules by individual intracellular vesicles (Wightman et al., 1991). The common practice in the analysis of amperometric spikes is to report their descriptive characteristics (such as peak current, width at half maximum, etc.). These characteristics are important since they encode information about how certain conditions and/or treatments affect exocytosis. However, analyzing the distributions of these descriptive characteristics does not provide answers to fundamental questions such as why a spike has a certain shape? What kind of physicochemical and biological processes underlie release events? How are different types of release related to the function of a cell under given conditions?

To answer these questions and related ones, theoretical models of exocytotic release must be available and applied. This would change the goal and practice of analyzing amperometric traces to provide additional insightful information about the relationship between various exocytotic modes through comparing their outcomes or predictions with experimental observations, depending on their different functions. However, to the best of our knowledge, there have been only a few attempts to establish such models without imposing ad hoc conditions on the nanoscale events or parameters assumed to be involved during exocytotic release or about the initial internal organization of the vesicle and their changes as exocytosis proceeds.

The purpose of this perspective is to expose several classes of amperometric spike types and to relate their morphological features to physicomathematical and biological descriptors. The principles considered hereafter are quite general and could be applied to various exocytotic systems,

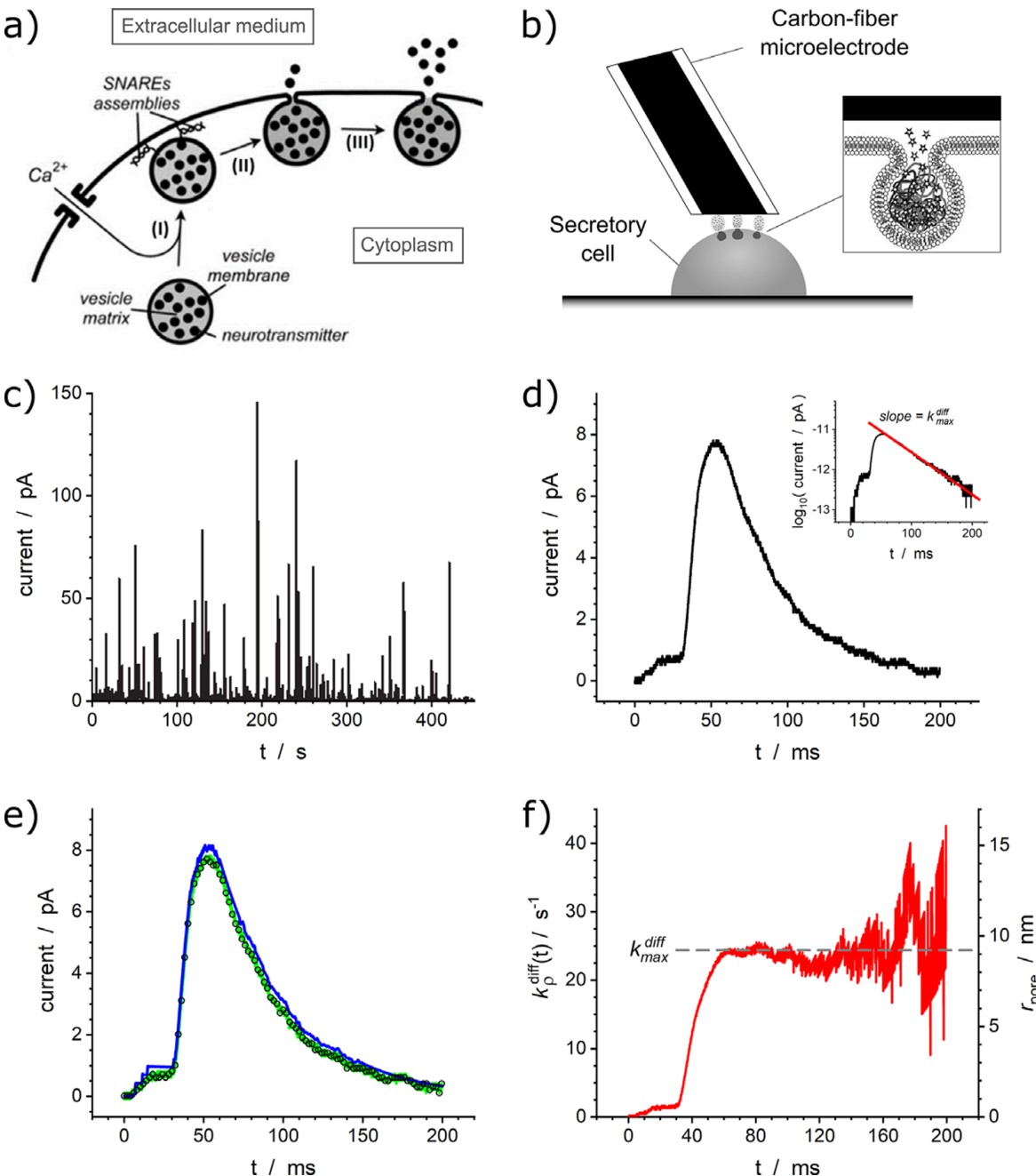

**Figure 1.** (*a,b*) Schematic representation of the sequential phases involved in a single vesicular exocytotic release event (*a*) and principle of its monitoring by single-cell amperometry (SCA) (*b*). (*c*) Typical SCA trace obtained from a bovine chromaffin cell (adapted from (Amatore et al., 2007a)). (*d*) Example of a spike with a single-exponential decay tail; the same spike is shown in the semi-log plot in the inset. (*e*) Simulations of the current spike shown in (*d*) using a 2D finite difference model with automatic iterative numerical tracking of the fusion pore opening rate (blue, adapted from (Amatore et al., 2010b)) or the comprehensive integral model discussed in this work (green, adapted from (Oleinick et al., 2017, Dannaoui et al., 2024)); open symbols reproduce the experimental spike. (*f*) Fusion pore opening dynamics extracted from the spike shown in (d) using the comprehensive integral model. In (*e*) and (*f*) $\kappa = 415\,\mathrm{s}^{-1}$ (see text).

although in the following we are mostly concerned with exocytosis from neuronal and neuroendocrine cells. The emphasis is placed on parsimonious models that accurately describe the physicochemical and structural properties that control the amperometric spikes' features and kinetics, while leading to relatively simple mathematical formulations that allow fast and precise numerical and statistical analyses of complete SCA traces. The presentation will be focused on two major axes: (i) characterization of the dynamics of the nanoscale

fusion pores through which neurotransmitters (NT) are released (Figure 1) and (ii) the effect of the nanoscale organization of intra-vesicular matrixes.

## Fusion pore dynamics

Regardless of the exact factors (i.e., proteic, lipidic or a mixture thereof) that determine the fusion pores structure and dynamics,

which are still under debate (Sharma and Lindau, 2018; Borges et al., 2023), fusion pores dynamics are formally mirrored by the time-dependent released NT fluxes, which are directly translated into amperometric spike currents owing to the SCA high temporal resolution. Basically, i.e., in physicochemical terms, these fluxes depend at any moment on the pore size, the mechanism of NT transport across the vesicle matrix and to the electrode surface, and the concentration of releasable NT remaining inside spherical vesicles, as will be illustrated for several spike types obtained from different cell types.

Storage of NT monocations in vesicles involves their chelation by polyelectrolyte matrixes (see below), so classical diffusion cannot occur. However, transport can still occur by kinetic exchange between occupied and unoccupied chelating sites. At a given constant temperature, such site-hopping diffusion is indistinguishable from normal diffusion (Ruff and Friedrich, 1971; Oleinick et al., 2017). The corresponding equivalent diffusion coefficients, $D_{ves}$, involve NT transfer across activation barriers between adjacent sites, so they are necessarily much smaller (ca. 1–2 orders of magnitude) than those, $D_{sol}$, in extracellular cell-electrode gaps (Amatore et al., 2000b). Since in a typical SCA experiment these gap widths are in the same nanometric ranges as the vesicles' sizes, it follows that the NT transport from the pore exit to the electrode can be neglected under most experimental conditions. Note that the condition $D_{ves} \ll D_{sol}$ is generally met so that the exact organization of extracellular space (e.g., glycocalyx) can be ignored, yet its effects should be somehow taken into account if the condition does not hold.

Finally, it is assumed that the electrode is biased at a sufficiently large overpotential to allow a quantitative collection by NT diffusion-limited oxidation. Under these conditions, the current monitored at the electrode during the exocytotic event is:

$$i(t) = nF J_{ves}^{tot}(t) \qquad (1)$$

where $n$ is the number of electrons transferred per neurotransmitter molecule, $F$ is the Faraday constant and $J_{ves}^{tot}(t)$ is the total NT flux released through the fusion pore at time $t$ since the opening of the fusion pore.

### Spikes with single exponential decay

Single-exponential spikes are very common and can be found in SCA traces recorded from a wide variety of cell types. Moreover, these spikes are important from a methodological point of view (see below) and represent the first category to have been rigorously rationalized historically (Amatore et al., 2000b, 2010a, 2010b; Oleinick et al., 2017).

The simple-exponential nature of spikes decay tails can be easily characterized by plotting the spikes currents in semi-log plots (see inset in Figure 1d) (Segura et al., 2000; Amatore et al., 2010a, 2010b; Oleinick et al., 2017; Borges et al., 2023). Their observations directly confirm the validity of the above assumptions about NT transport and the constant size of the fusion pore (Amatore et al., 2010a, 2010b). This is indeed in accordance with Newton's or Kelvin's laws, which were established for heat diffusion during cooling of solid bodies, even if in the present case diffusion occurs through site-hopping NT exchanges. The ratio of the maximum fusion pore radius, $r_{pore}^{max}$, to that of the vesicle, $r_{ves}$, can then be directly deduced from the slope, $k_{max}^{diff}$, of the linear fit of the decay tail represented in a semi-log-plot (see inset in Figure 1d) (Amatore et al., 2010b; Oleinick et al., 2013; Oleinick et al., 2017; Dannaoui et al., 2024):

$$r_{pore}^{max}/r_{ves} = k_{max}^{diff}/\kappa \qquad (2)$$

where $\kappa = D_{ves}/r_{ves}^2$ is the rate of NT equivalent diffusion in the vesicle.

Precise 2D numerical simulations (Amatore et al., 2010a, 2010b) established that this property results from the establishment of a quasi-steady state diffusion regime inside the vesicle once the release time duration exceeds $t_{min} = 1/\kappa = r_{ves}^2/D_{ves}$ (Amatore et al., 2010a, 2010b; Oleinick et al., 2017). When this regime is reached, the NT concentration map inside the vesicle during release decreases its amplitude with time while maintaining its shape homothetically. Therefore, once the fusion pore radius becomes constant, the NT release flux from the vesicle is directly proportional to the total amount of NT remaining inside the vesicle (see (Oleinick et al., 2017) and its supporting information), which is equivalent to saying that the release flux varies exponentially with time. It is important to note that for almost all amperometric spikes, the value of $t_{min}$ value is much smaller than the time at which the exponential regime begins. This implies that the global shape of current spikes directly reflects the time variations of fusion pore size even before the pre-exponential phase, i.e., during the rising and spike maximum regions, as well as during the development of any possible pre-spike features. This critical finding (Oleinick et al., 2017) has been tested using 2D numerical simulations (Amatore et al., 2010b) and found to be perfectly correct with an accuracy and precision better than that of amperometric current spikes measurements (see, e.g., Figure 1e). This result is significant because it transforms a complex problem into a simple one. Rather than solving a time-dependent 2D partial derivative equation with a moving boundary that requires input of precise dimensions as well as iterative adjustments to track a priori unknown pore dynamics (as was done in Amatore et al., 2010b and Oleinick et al., 2013), our present approach led to using a simple ordinary differential equation (Oleinick et al., 2017). The latter approach rooted on Lord Kelvin's modeling of heat release from a body naturally provides the fusion pore opening kinetics without making any assumptions. Figure 1e, which compares the result of the 2D approach with that of the comprehensive integral approach when simulating the current spike shown in Figure 1d, shows the spectacular performance of the latter, especially considering the drastic reduction of computation time and memory requirements. Most importantly, this novel compact approach allows one to derive several closed-form analytical results, including a direct analytical relationship between the spike current, $i(t)$, and the time-dependent radius of the fusion pore, $r_{pore}(t)$ (Oleinick et al., 2017; Dannaoui et al., 2024):

$$\frac{i(t)}{Q_0 - \int_0^t i(\theta)\,d\theta} = \left(\frac{D_{ves}}{r_{ves}^2}\frac{r_{pore}^{max}}{r_{ves}}\right)\frac{r_{pore}(t)}{r_{pore}^{max}} = k_{max}^{diff} \times \frac{r_{pore}(t)}{r_{pore}^{max}} = k_\rho^{diff}(t) \quad (3)$$

where $r_{pore}(t)$ is the time-dependent pore radius ($\rho = r_{pore}(t)/r_{pore}^{max}$), $k_\rho^{diff}(t)$ is a pseudo-rate constant featuring the NT release kinetics at each moment, and $Q_0 = nF q_0$, where $q_0$ is the total releasable content of NT in the vesicle (number of moles), and $\theta$ is an integration variable. Note that we introduce here the term "releasable content," as opposed to the total content, because it characterizes the NT amount which is released during the whole duration of amperometric spikes as obtained through a simple time integration of the spike current and not necessarily the total storage content of the vesicle before release (see below). Figure 1f shows

the variations of $k_\rho^{\text{diff}}(t)$ with time, as well as those of the corresponding fusion pore radius, obtained by applying Eq. (3) to the experimental current spike shown Figure 1d. Note that in Figure 1f, the red horizontal line is shown to indicate the average pore radius for $t \geq 0.07\,\text{s}$ since the sensitivity of Eq. (3) is so high that it can drastically amplify the current noise when both $i(t)$ and $Q_0 - \int_0^t i(\theta)\,d\theta$ values approach zero. This may be the cause of the rapid oscillations observed in this time range (compare Figure 2Ia,b or those in (Oleinick et al., 2017), for cases involving much lower current noise).

This comprehensive integral model corresponds perfectly to the notion of "parsimonious models," viz., of models that provide the desired level of explanation or prediction while relying on as few predictor variables as possible. Indeed, while being based on a solid physicochemical and biological basis, it remains simple and accurate. Moreover, its analytical character (see Eq. (3)) makes it perfectly suited to process large data sets in sufficiently short times to be of interest to experimentalists during their measurements or immediately afterward. For example, using this parsimonious model, it was possible to analyze whole SCA traces recorded under different experimental conditions at chromaffin cells to derive sound statistical data about the dynamics of fusion pores opening and how they responded to these conditions (Figure 2a) (Oleinick et al., 2017). In particular, these analyses showed that for chromaffin or PC12 cells, the fusion pore sizes remain considerably smaller than those of the vesicle over the whole duration of amperometric spikes.

Another major advantage of such parsimonious model is that it can be easily adapted to various experimental situations without the need to specify the exact nature of the species stored inside the vesicles or the geometry of the releasing vesicles a priori (see, e.g., an application to ROS homeostatic release (Zhang et al., 2019), as would be required by conventional 2D finite difference or finite element numerical approaches. In addition, the reconstructed fusion pore dynamics naturally reveal pore closures (temporal or definitive) if this happens during release event. For example, when exocytotic release is directly monitored by a nanoelectrode inserted inside a self-reconstructed rat functional neuromuscular junction (NMJ) (Li et al., 2015a), a sequence of brief 400 Hz periodic current spikes is clearly observed (Figure 2IIa), strongly suggesting the involvement of successive opening/closing phases of the fusion pore as confirmed in Figure 2IIb, which represents the corresponding fusion pore fluctuations. Note that the maximum pore radius is ca. 1.2 nm, in perfect agreement with the estimate of initial fusion pore dimensions independently characterized by patch-clamp measurements (Albillos et al., 1997). The entire sequence in Figure 2IIa was treated as a whole. Thus, it is remarkable that the decreasing amplitude of each successive current peak was fully consistent with the progressive loss of ca. 7,000 NT molecules from the vesicle during each opening phase, even though the maximum fusion pore radius was found to be identical during each opening, without any a priori constraint (Li et al., 2015a).

The same parsimonious model can equally be applied to a situation where exocytotic release is monitored by a micro−/nanoelectrode placed in close proximity to a functional NMJ of a Drosophila larva (Majdi et al., 2015; Larsson et al., 2020). Figure 2IIIa,b shows the observed sequence of current spikes and the extracted dynamics of the fusion pore in this case. It can be seen that the extracted fusion pore opening/closing sequence tracks extremely well the experimental current fluctuations and again yields a maximum size of ca. 1.2 nm for the fusion pore radius.

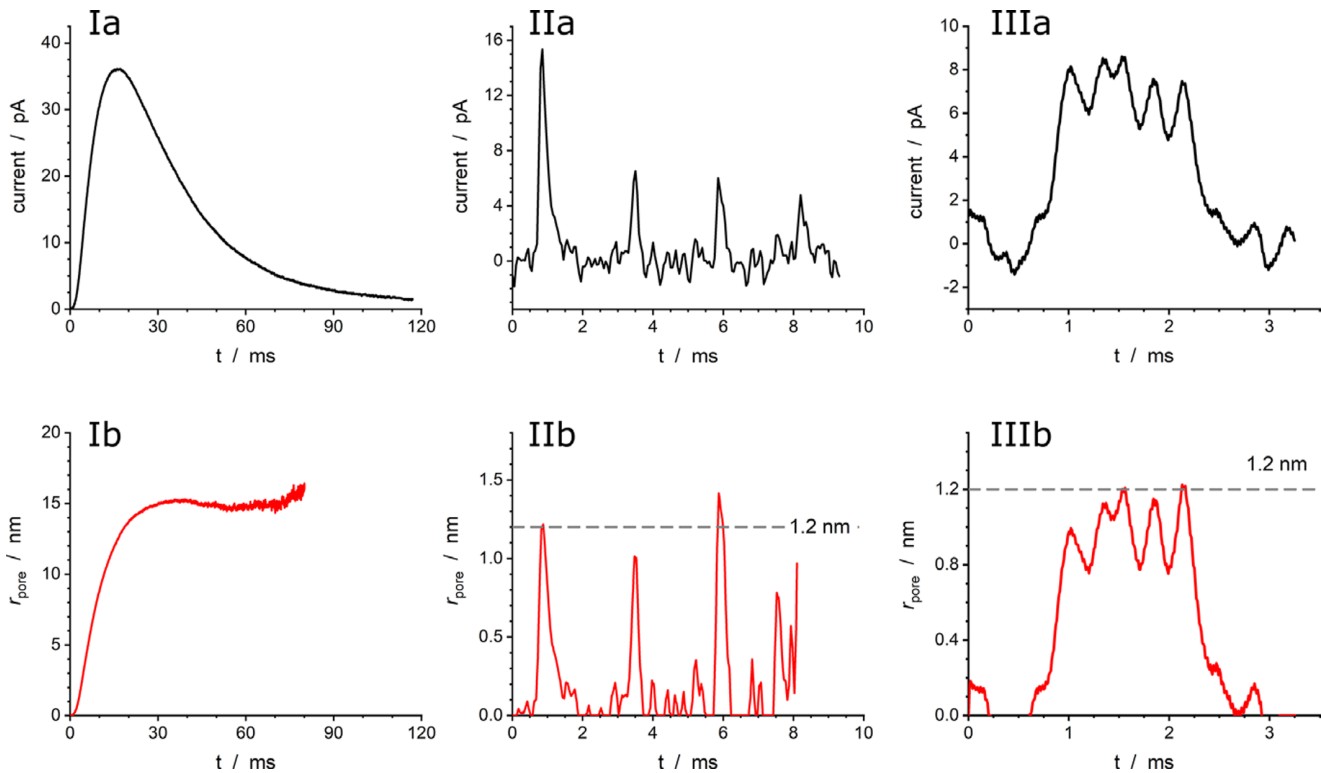

**Figure 2.** (a) Experimental current spikes and (b) corresponding fusion pore dynamics: (I) Single amperometric spike from bovine chromaffin cell (Oleinick et al., 2017); intrasynaptic complex release sequences monitored (II) inside a self-reconstructed rat neuro-muscular junction (Li et al., 2015a) or (III) in the vicinity of neuromuscular junction of Drosophila larva (Larsson et al., 2020).

Note, however, that, compared to the intrasynaptic case in Figure 2IIa,b, it is difficult to conclude if these sequences involve a series of complete closing phases, because diffusion from the pore opening to the electrode surface may smooth sharp transitions (Schroeder et al., 1992) due to the extra-synaptic placement of the electrode tip.

## Intravesicular organization

The previous section considered spikes exhibiting single-exponential decay tails as well as some other events that could be treated with the same formalism. Although most published reports implicitly assume that all amperometric spikes follow such behavior, a non-negligible proportion of them clearly exhibits non-exponential decay tails with a second exponential regime characterized by a smaller slope than the first one. This type of situation is readily and conveniently identified by displaying the current spikes in semi-log-scale plots (Figure 3a,b). Relying on Eq. (2) or its previous intuitive equivalents, it is often argued that the second, slower exponential release features a decrease in the fusion pore radius after having reached its maximum size, which corresponds to the first exponential regime.

However, except when the perfectly recognizable features of a complete fusion pore closure event are observed as in Figure 2II,III, or reported for very few other cases (Mellander et al., 2012), such a rationalization is inconsistent with the physics of exocytotic vesicular release. Indeed, upon considering that the second exponential regime has a smaller slope (Figure 3d) due to a smaller pore size, one would necessarily arrive at the paradoxical situation that a smaller fusion pore, e.g., as illustrated in Figure 3c, would nevertheless allow a larger release NT flux than predicted by extrapolating the first-exponential (see Eqs. (1–2)) (Mellander et al., 2012). Figure 3b, where the orange dashed line shows the extrapolated current decay predicted if the pore had retained its maximum size leading to the first exponential regime, clearly illustrates the inconsistency of this explanation.

This is better explained and confirmed by the simulation results displayed in Figure 3d when a fusion pore decreases its size during a second phase of the release event (Figure 3c). As expected, such pore size diminution necessarily results in a characteristic current dip when the pore size starts to decrease (Figure 3d). Only after this fall ends, i.e., when the fusion pore reaches its smaller size, the current may decay with a smaller exponential rate (inset in Figure 3d), leading to a later crossing above the extrapolated current

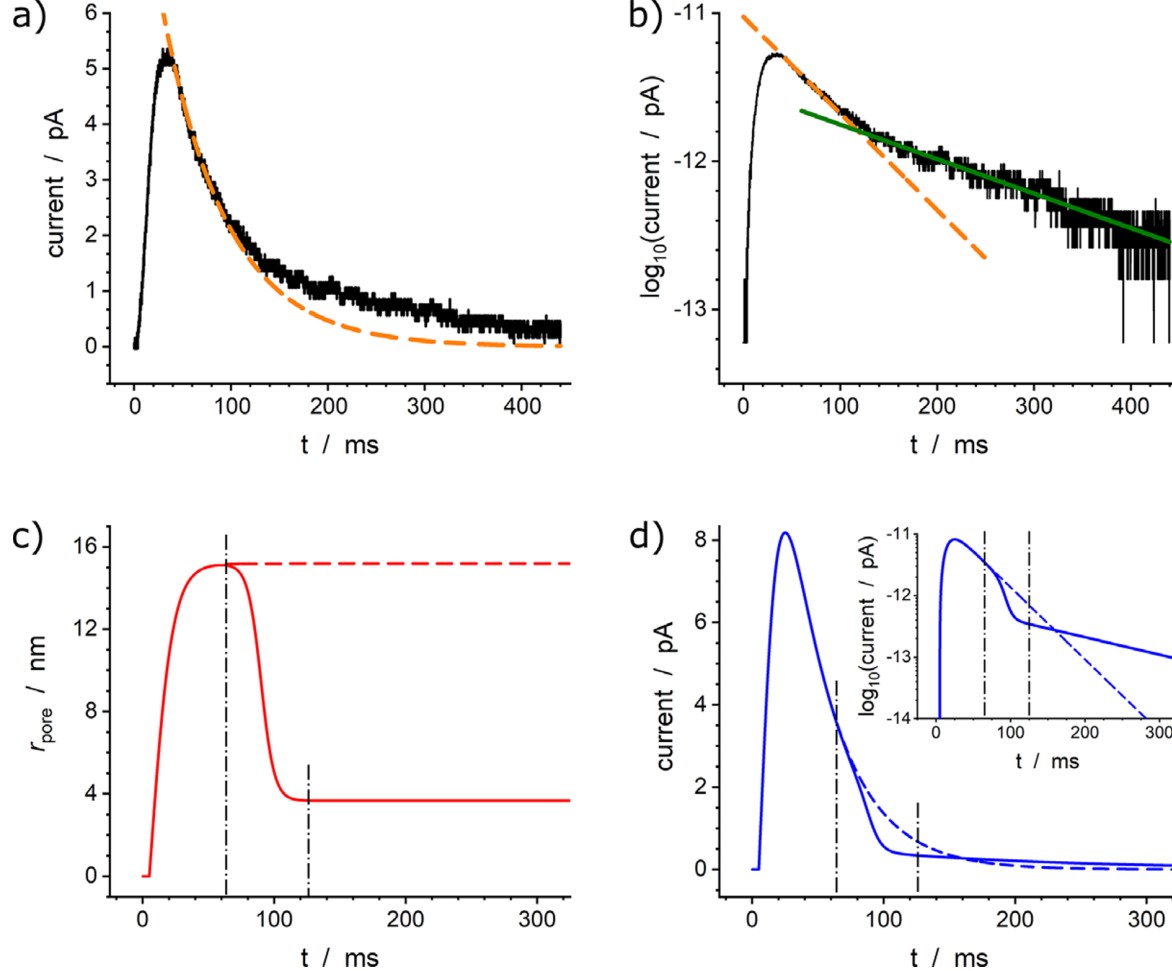

**Figure 3.** Single amperometric spike from a chromaffin cell exhibiting a two-exponential decay tail in real (*a*) or semi-log-scales (*b*); in (*a, b*) the extrapolated first-exponential decay is superimposed as the orange dashed curve (*a*) or line (*b*), while in (*b*) the second exponential limit is superimposed as the solid green line; compare Figure 4*b–c*. (*c*) Schematic illustration of a fusion pore dynamics with an opening phase followed by a partial closure (solid curve) or stalled at its maximum size (dashed horizontal line); (*d*) corresponding amperometric spike (blue solid) in real time scale together with the extrapolation of the first exponential mode (blue dashed curve); inset in (*d*) shows the same amperometric spike modes in semi-log-scale.

expected if the fusion pore had kept its size (blue dashed curve in Figure 3d). Nonetheless, this cannot happen until long after the end of decay phase of the fusion pore. However, to the best of our knowledge, this situation is never observed for experimental spikes with two exponential decay tails such as that in Figure 3a, since the change in release rate always occurs without showing any transient current drop (compare green and orange curves in Figure 3b).

This led to the proposal (Oleinick et al., 2016; Hu et al., 2016; Dannaoui et al., 2024) of a more physically realistic rationale based

on soft matter physics, taking into account the biological fact that the high intravesicular storage of NT cations in large dense core vesicles of neuroendocrine cells mostly occurs through chelation by negatively charged polyelectrolyte proteins of the chromogranin family (Borges et al., 2010). Condensed polyelectrolytes in the presence of cationic NT (e.g., catecholamines), prone to establish a dense network of H-bonds, necessarily adopt a biphasic structure (Figure 4a) (Oleinick et al., 2016; Dannaoui et al., 2024). To the best of our knowledge, the presence of 'blobs' and 'strands' in secretory

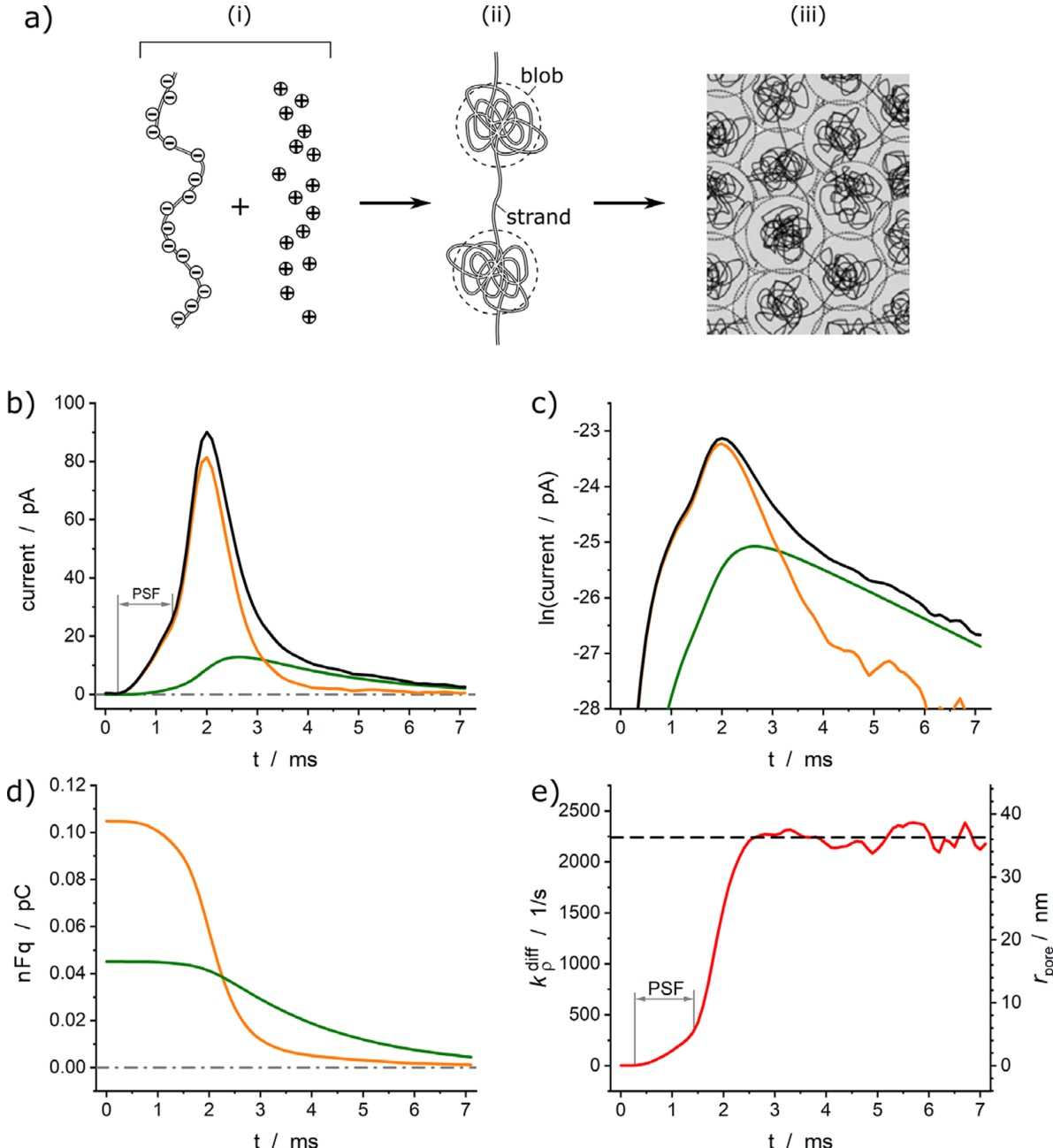

**Figure 4.** (a) Principle of the formation of a biphasic matrix structure inside a vesicle: (i) schematic representation of polyelectrolyte facing a pool of NT cations; (ii) electrostatic condensation of the system in (i) after entropic reorganization; (iii) final structure of the system in (ii) when compacted inside of a vesicle. (b) Experimental and simulated amperometric spikes (black) exhibiting a two-exponential decay tail from a PC12 cell superimposed to its two components featuring the current contributions from each domain (orange: less compacted phase; green: highly compacted phase); (c) same as (b) plotted in semi-log-scale; (d) corresponding variations of releasable NT quantities in each sub-domain shown in terms of the stored charges ($n = 2$); (e) fusion pore dynamics in terms of the apparent rate constant $k_\rho^{diff}(t)$ (Eq. (3), left vertical axis) and corresponding fusion pore radius (right vertical axis, for $r_{ves} = 94$ nm). $\kappa = 5803 \, s^{-1}$ and $k_1 q_{lc}^0 = 470 \, s^{-1}$; $q_{hc}^0 = 0.55$ amol and $q_{lc}^0 = 0.23$ amol are the optimized initial NT releasable contents in the highly and less compacted domains (adapted from (Dannaoui et al., 2024)).

vesicles has not yet been confirmed by microscopy, although their formation is well established in the field of polyelectrolytes (see, e.g., Wittmer et al., 1995; Muthukumar, 2004).

The sizes of the 'blobs' and 'strands' sketched in Figure 4a-(ii) are imposed by entropic requirements since a fully compacted monophase would result in too high a negative entropy (Oleinick et al., 2016). The folding of the blob-strand structures depicted in Figure 4a-(ii) within vesicles, necessary leads to the formation of highly condensed grainy blobs dispersed in a less-condensed phase formed by the poorly compacted strands (Figure 4a-(iii)). Prior to exocytotic release, NT cations are dispersed in equilibrium between these two phases, being more stabilized in the most compacted domains because of their multiple interactions (electrostatic, H-bonds, etc.) with the polyelectrolyte backbone moieties. This hampers their diffusion by site-hopping. Conversely, those in the less compacted domain are prone to diffuse faster by site-hopping to be easily released into the extracellular medium as soon as the fusion pore opens (compare Figure 3 in (Ren et al., 2020)).

The complex release dynamics may then be described as involving a transfer of NT cations, Eq. (4a), from the highly condensed phase to the less condensed one, slowly replacing part of those which were already been released through the site-hopping process, Eq. (4b):

$$q_{\text{hc}} + \varnothing_{\text{lc}} \xrightarrow{k_1} \varnothing_{\text{hc}} + q_{\text{lc}} \tag{4a}$$

$$q_{\text{lc}} \xrightarrow{k_\rho^{\text{diff}}} q_{\text{out}} \tag{4b}$$

where $q_{\text{hc}}$ and $q_{\text{lc}}$ are the time-dependent quantities of the total releasable neurotransmitter in the highly compacted (hc) and less compacted (lc) domains, respectively; $\varnothing_{\text{hc}}$, $\varnothing_{\text{lc}}$ are the quantities of chromogranin 'free' sites (i.e., where NT cations have been substituted by other cations not prone to be strongly chelated by the polyelectrolyte backbone as efficiently as NT cations) in each domain; $k_1$ is the bimolecular rate constant characterizing the global exchange of NT in Eq. (4a); $k_\rho^{\text{diff}}(t)$ is the time-dependent apparent rate constant governing the NT release in the vesicle-electrode gap and is proportional to the time-dependent fusion pore radius, and whose maximum value is $k_{max}^{\text{diff}}$ (see Eqs. (2) and (3)).

According to thermodynamics, an opposite displacement of NT cations toward the more condensed domains should occur in an attempt to restore the initial equilibrium in Eq. (4a). However, this process is likely very slow due to the high activation barriers of the corresponding NT transfers and the negligible number of free sites, $\varnothing_{\text{hc}}$, in the most condensed domains at the beginning of the release. Given the rapid rate of release into the extracellular gap in Eq. (4b), the backward reaction (4a) may be rightly assumed to play a negligible kinetic role during most of the release process. Conversely, the forward reaction (4a) may start to operate as soon as the proportion of free sites $\varnothing_{lc}$ in the less condensed phase becomes significant (compare Figure 4d).

Although to the best of our knowledge, such a case has never been reported, the above formalism can easily be extended to situations involving more than two exponential regimes by invoking the existence of more types of compacted domains as sketched in *Figure 4a(iii)*. Indeed, the present model can be easily extended by considering an arbitrary number of domains of various compactness and introducing a corresponding number of additional equations, such as Eq. (4a), which intervene in a cascade and/or

sequence, as performed in a different context (Oleinick et al., 2018). However, since we have never encountered such situations, for the sake of clarity and simplicity, we limited our presentation to the consideration of two domains of different compactness which, by the way, covers the "dense core and halo" case often discussed in the literature, whatever is the exact physicochemical nature of the light areas named "halos" observed in EM. Although the exact chemical and physicochemical nature of these storage domains are still a matter of debate (Borges et al., 2010), by non-specifying the exact nature of the interactions between the matrixes and the neurotransmitters, the model naturally encompasses also the situation when the transmitter is not bound to chromogranin in the vesicle "halo."

As for the monophasic vesicles, this model can be formalized as a system of two ordinary differential equations giving rise to spikes with two-exponential decay tails while the fusion pore keeps its maximum size. $k_1$ and $k_{\text{max}}^{\text{diff}}$ values can be derived from those of the experimental slopes or each exponential component (Dannaoui et al., 2024). Also, the fusion pore dynamics, including when a pre-spike feature (foot) is present (Amatore et al., 2007b), can be easily extracted applying Eq. (3) to the experimental spike current before the first exponential component is observed. Figure 4b–e provides an example of this procedure applied to a typical spike with PSF recorded at a PC12 cell (Dannaoui et al., 2024). It should be noted that this model is intended to describe the spikes having two well-defined exponential regimes (see also a discussion about the kinetic coupling between the beginning of the first exponential regime and the end of the fusion pore enlargement in Dannaoui et al., 2024). Although we have never observed such cases, if other features were to interfere with the beginning of the first exponential regime near the spike maximum, such as a trend toward a small decrease of the fusion pore radius, they would require alternative rationalizations. Nonetheless, in the case of single exponential tails, the analytical expression of the fusion pore expansion kinetics (Eq. 3), none of the ensuing pore dynamics suggested that such a case may exist.

Interestingly, the biphasic matrix structure allows rationalizing the reported dichotomy between sub-quantal and quantal release, or in other terms between partial and full fusion (Ren et al., 2016), without the need to invoke a closing fusion pore (this is evidently applied to the spikes not possessing clear closing signature, examples of such signatures are shown in Figures 2IIa, 2-IIIa and 3d or, e.g., in supporting information of (Mellander et al., 2012). Indeed, the intravesicular matrix must spontaneously swell when NT cations of the compacted domains are replaced by ions present in the extracellular fluid to maintain electroneutrality without leading to strong chelation of the replacement ions (Marszalek et al., 1997; Tanaka and Fillmore, 1979). However, while the vesicle membrane remains intact – except for the small fusion pore area that amounts to only a few hundredth percent of its surface area – the matrix cannot swell completely (Marszalek et al., 1997; Amatore et al., 2000a; Amatore et al., 2000b) to release its full NT content. In other words, during an amperometric spike, a significant fraction of the initial NT content cannot be released due to the membrane restricting the extent of matrix swelling and hence hindering the exchange of chelated NT cations. This is equivalent to saying that the NT partition coefficient towards the extravesicular medium tends to become negligible before the matrix is fully emptied. When this happens, the released flux is expected to be too small to be detected amperometrically, giving the appearance of subquantal release. This view is coherent with recent STED observations of the vesicle fate after release (Shin et al., 2020). Although the time

resolution of those measurements was not enough to characterize the release along the duration of the amperometric spike. However, in the post-spike phase, the pore may be closed (without leaving a clear signature in the current signal due to its amplitude comparable to noise) or further enlarged, leading to release of whole vesicle cargo, including chromogranin matrix. In the latter case, the fusion pore expands to reach a size comparable to that of the vesicle, as in full fusion (Harata et al., 2006; Verdugo, 1993) or during FVEC/VIEC/IVIEC[1] experiments (Dunevall et al., 2017; Li et al., 2015b), swelling will no longer be refrained, and a quantal release should be observed.

In addition, NanoSIMS measurements (Nguyen et al., 2023) show a decrease of the transmitter concentration within the dense core after vesicle partial release, which supports the views considered here. The observations that release fraction does not depend on vesicle size, as well as a decrease of the vesicle size (while dense core maintaining its original size) after exocytotic release, are coherent with the active role of intravesicular matrix in controlling release as formalized by the models discussed in this work.

## Outlook

This work summarizes the physico-mathematical parsimonious models developed in our group to understand exocytotic vesicular release events monitored by amperometry. Instead of characterizing the shape of amperometric spikes using only descriptive parameters, these models aim to identify the primary biological and physicochemical factors that determine the shape and intensity of these spikes, while also allowing for the extraction of structural and kinetic parameters that define them using direct analytical or fitting procedures.

These comprehensive models were designed to be parsimonious. That is, while they reflect the main underlying biological and physicochemical phenomena, they remain simple, accurate, and computationally inexpensive in terms of memory usage and execution speed compared to more classical approaches relying on complex numerical simulations based on time-dependent 2D or 3D finite differences or finite elements coupled with time-tracking of unknown fusion pore size dynamics. This parsimony is essential for fast and possibly automatic processing of large data sets corresponding to hundreds of spikes present in amperometric traces. This is indeed required for providing statistically significant analyses to reveal and establish key biological patterns, which are the main purpose of these analytical measurements. Additionally, these comprehensive models offer a means of predicting complex experimental outcomes and facilitate comparisons between different cells (PC12, bovine chromaffin, neuromuscular synaptic junctions, as reported here or for other systems published elsewhere by our group or others) investigated by different methods, whether amperometric or not, providing a deeper understanding and more accurate description of single vesicular exocytotic events.

**Open peer review.** To view the open peer review materials for this article, please visit http://doi.org/10.1017/qrd.2025.10010.

**Data availability statement.** No new data was generated during this work.

---

[1]The abbreviations mean flow vesicle electrochemical cytometry, vesicle impact electrochemical cytometry, intracellular vesicle impact electrochemical cytometry, correspondingly.

**Acknowledgements.** We wish to specially thank all our collaborators whose names appear in the references for their crucial involvement in the published works that this perspective summarizes.

**Financial support.** This work was supported in part by CNRS, ENS, PSL University, and Sorbonne University (UMR 8228 CPCV). CA thanks Xiamen University as well for his Distinguished Visiting Professor position.

**Competing interests.** The authors declare no conflict of interests.

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
