## [Reviewer Report]

This Perspective article from Amatore and colleagues presents a clear and logical summary of the contributions of mathematical modeling to the interpretation of experimental observations of single exocytotic events with high-resolution single-cell amperometry. Two key insights emerge: the importance of the dynamics of the fusion pore that connects the inner volume of the vesicle to the extracellular space, and the interactions of cationic neurotransmitter molecules with the polyelectrolyte chromogranin proteins contained with the vesicles. The models of these factors describe and rationalize the key features of amperometric spikes exhibiting mono- and bi-exponential decay profiles.

---

## [Reviewer Report]

This is a highly interesting perspective on the work done by these authors to model the release of neurotransmitter molecules via exocytosis. The work provides insights and discussion of what might be occurring during the process of emptying a vesicle. Although this reviewer does not agree with all the discussion, the general themes are in agreement and the work is very valuable.

The following is a list of comments for consideration.

Page 2, it seems the word “review” should be “perspective” as this is not really a review, but presentation of some important models with speculation of their application and meaning.

Is it clearly established that all the neurotransmitter molecules in the vesicle are bound to matrix? It seems molecules that are not bound need to be accounted for somehow.

Is there an impact of the extracellular matrix on diffusion when modeling release in the space between the cell membrane and the electrode?

Page 5, the sentence beginning with “This is an important result” is very long and confusing. Is it possible to break this into concrete units?

Page 8, there might be an alternative explanation for the two exponential decays where the second is above the expected single decay trace. The falling part of the spike could be defined, in part, by the pore closing and the later or second part of the trace is larger than expected owing to a ‘slowing' of the rate of pore closing. This would lead to the slower decay observed. Has this been considered here?

Page 9, this following is written very forcefully, “biological fact that the high intravesicular storage of NT cations in large dense core vesicles of neuroendocrine cells mostly occurs through chelation by negatively charged polyelectrolyte proteins of the chromogranin family”. In fact, Borges et al, stated the larger portion of transmitter was bound to matrix and this was not definitive in that paper. The existence of bound and and unbound transmitter in vesicles is still under quite some debate and this is discussed in Borges et al 2023. This should be mentioned and considered.

The “blob-strand” model is fascinating speculation, and, this reviewer understands QRB-D allows a great deal of speculation, but the authors should include or mention any other alternatives that are considered realistic. The discussion of condensed domains on page 9 could also be observed with a vesicular system comprising a protein dense core to bind some of the neurotransmitter molecules and a fluid halo in which diffusion would be more rapid. Even in fast frozen samples these halos are observed and they increase and decrease following transmitter level changes (pharma) when observed in fixed tissues, implying they are not artifacts.

Is there observational evidence for the presence of blobs and strands in the vesicular matrix? If not, this model is still intriguing, but this should be stated.

The paragraph beginning with, “Interestingly, the biphasic matrix structure allows rationalizing the reported dichotomy between subquantal and quantal release without the need to invoke a closing fusion pore.” Is in need of clarification. Can the authors clarify the mechanism they propose for sub-quantal release and the fusion pore remaining open? Can the authors clarify with a mechanism to explain how matrix swelling results in incomplete release of transmitter? The argument that the amount of transmitter becomes too low to be observed does not appear to hold for systems that release only 5 or 10% of the cargo - as observed in the 2020 Larson et al paper for example (on which the authors here were coauthors). This should be clarified.

Also, how does one define quantal? Sub quantal seems to be an odd term. The literature, including many papers by these authors, seems to use partial release to describe the phenomenon.

Are FVEC, VIEC and IVIEC defined anywhere?

---

## [Reviewer Report]

The authors seem to be defensive, argumentative, and even sarcastic when my comments were meant to request clarification and be helpful. I think some questions were not answered and could easily be done with little effort. I have looked over the revised manuscript and the responses and list further comments to 2, 5, 6, 7, 9.

This is a highly interesting perspective on the work done by these authors to model the release of neurotransmitter molecules via exocytosis. The work provides insights and discussion of what might be occurring during the process of emptying a vesicle. Although this Reviewer does not agree with all the discussion, the general themes are in agreement and the work is very valuable.

The following is a list of comments for consideration.

We thank the Reviewer for this general appreciation of our work and we wish to address the raised comments one by one below.

2. Is it clearly established that all the neurotransmitter molecules in the vesicle are bound to matrix? It seems molecules that are not bound need to be accounted for somehow.

This point is still a matter of debate as indicated by the Reviewer in her/his comment number 6 below. Thus, we address this question when answering this later point.

Reviewer response: The point is that the manuscript at written is not clear on this point (see below).

5. Page 8, there might be an alternative explanation for the two exponential decays where the second is above the expected single decay trace. The falling part of the spike could be defined, in part, by the pore closing and the later or second part of the trace is larger than expected owing to a ‘slowing' of the rate of pore closing. This would lead to the slower decay observed. Has this been considered here?

If the falling part of the spike was to involve the pore closure kinetics in addition to the diffusion rate, its time variation could not be exponential unless the pore closing law was to accidentally couple with the neurotransmitter diffusion law in order to produce a global exponential decay. It is difficult to imagine that such a complex delicate coupling would persist systematically regardless of the cell. Therefore, this type of spikes is not within the scope of the present manuscript.

Reviewer response: The authors argue without clarification. The curves for the release events are fitted to best exponentials. This does not mean they are strictly exponential. This reviewer finds the argument lacking.

6. Page 9, this following is written very forcefully, “biological fact that the high intravesicular storage of NT cations in large dense core vesicles of neuroendocrine cells mostly occurs through chelation by negatively charged polyelectrolyte proteins of the chromogranin family”. In fact, Borges et al, stated the larger portion of transmitter was bound to matrix and this was not definitive in that paper. The existence of bound and unbound transmitter in vesicles is still under quite some debate and this is discussed in Borges et al 2023. This should be mentioned and considered.

After the sentence indicated by the Reviewer we cited 2010 Borges et al paper. In this paper Borges et al stated that chromogranins bind transmitter molecules in order to reduce osmotic pressure and increase storage capacity. Thus, our sentence is in agreement with the statement from the cited paper.

Reviewer response: The point here is that the statement in the manuscript is too strong, and Borges was first, not so forceful, and second, speculative. Thus, this sentence remains unclear.

For the second part of the comment (which also addresses the Reviewer’s comment 2), the model for two-exponential spikes explicitly defines two populations of neurotransmitter molecules, q_hc and q_lc, stored in the highly compacted and less compacted intravesicular domains (page 9). These populations are considered as neurotransmitter molecules that are strongly or weakly chelated by the polyelectrolyte, respectively, which is similar to the ‘bound’ and ‘unbound’ categories indicated by the Reviewer. Therefore, this point is already mentioned and considered in the manuscript.

Reviewer response: The authors seem to ignore the possibility that some parts of the vesicle might have aqueous solution that is not occupied by polyelectrolyte. The model seems to allow for this case, but the point to make is that the authors allow a bias by virtue of their wording to imply all transmitter molecules are bound. This is a simple thing to correct.

7. The “blob-strand” model is fascinating speculation, and, this Reviewer understands QRB-D allows a great deal of speculation, but the authors should include or mention any other alternatives that are considered realistic. The discussion of condensed domains on page 9 could also be observed with a vesicular system comprising a protein dense core to bind some of the neurotransmitter molecules and a fluid halo in which diffusion would be more rapid. Even in fast frozen samples these halos are observed and they increase and decrease following transmitter level changes (pharma) when observed in fixed tissues, implying they are not artifacts.

The model for two-exponential spikes, as discussed in the manuscript, is not limited to a fixed number of dense domains. Thus, as the Reviewer correctly pointed out, it can also be applied to a single dense core and halo configuration.

According to classical soft matter physics of polyelectrolytes, it is clear that when the most stable domain (i.e., the most condensed one) is fully loaded, new material imported into (or released from) the vesicle necessarily accumulates in (or originates from, respectively) the less condensed domain. Therefore, except for the use of the poetic but imprecise notion of a “fluid halo” to characterize the less condensed domain, the Reviewer’s description and reported experiments are fully coherent with our model.

Reviewer response: The unnecessary sarcasm aside, the authors do now at least add, “we limited our presentation to two domains of different compactness which, by the way, covers the “dense core and halo” case often discussed in the literature.”

This new statement does not befit the authors sense of being comprehensive and considering the work of others. It would be better to reword this to be less biased in the inference and superiority.

9. The paragraph beginning with, “Interestingly, the biphasic matrix structure allows rationalizing the reported dichotomy between subquantal and quantal release without the need to invoke a closing fusion pore.” Is in need of clarification. Can the authors clarify the mechanism they propose for sub-quantal release and the fusion pore remaining open? Can the authors clarify with a mechanism to explain how matrix swelling results in incomplete release of transmitter? The argument that the amount of transmitter becomes too low to be observed does not appear to hold for systems that release only 5 or 10% of the cargo - as observed in the 2020 Larson et al paper for example (on which the authors here were coauthors). This should be clarified.

The last paragraph of the ‘Intravesicular organization’ section (page 11) fully explains the mechanistic possibility envisioned by the authors. It reads as follows:

“Interestingly, the biphasic matrix structure allows rationalizing the reported dichotomy between sub-quantal and quantal release, or in other terms between partial and full fusion (Ren et al., 2016) without the need to invoke a closing fusion pore. Indeed, the intravesicular matrix must spontaneously swell when NT cations of the compacted domains are replaced by ions present in the extracellular fluid to maintain electroneutrality without leading to strong chelation (Marszalek et al., 1997; Tanaka and Fillmore, 1979). However, while the vesicle membrane remains intact — except for the small fusion pore area that amounts to only a few hundredth percent of its surface area — the matrix cannot swell completely (Marszalek et al., 1997; Amatore et al., 2000b) to release its full NT content. In other words, during most of NT release, a significant fraction of the initial NT content cannot be released due to the membrane restricting the extent of matrix swelling and hence the exchange of chelated NT cations. This is equivalent to saying that the NT partition coefficient towards the extravesicular medium tends to become negligible before the matrix is fully emptied. When this happens, the released flux is expected to be too small to be detected amperometrically, giving the appearance of subquantal release. However, if the fusion pore expands to reach a size comparable to that of the vesicle before this happens, as in full fusion (Harata et al., 2006) or during FVEC/VIEC/IVIEC experiments (Dunevall et al., 2017; Li et al., 2015b), swelling will no longer be refrained, and a quantal release should be observed.”

This is consistent with the sub-quantal release observed in Drosophila larval neurons, as reported in the Larsson et al. (2020) paper (note that Larsson’s name takes two ‘s’). Additionally, we must point out that in this paper, the opening/closing succession of phases involved fusion pores with a maximum radius of approximately 1.2 nm or less (see Figs 2IIIa & 2IIIb). In other words, these release events were not related to fusion pores that had expanded from their initial stage. Rather, they were related to fusion pores associated with the classical PSF features observed in amperometric monitoring. Therefore, it is not surprising that only small quantities were released, and the Reviewer’s remark cannot be used to disprove any of the conclusions of our model.

Reviewer response: No effort was made to clarify the paragraph in question - which might be clear to the authors but will likely not be clear to readers. This reviewer finds the paragraph and the response questionable without further clarification. The entire paragraph is highly speculative (which is good, but not clear), and ignores other literature that is now out where vesicles have been imaged (both by STED and NanoSIMS, the first looking at details of the pore closing and the second looking at relative levels of neurotransmitter in vesicles before and after release). I simply do not know why the authors choose to go on this limb without a clearer explanation.

Also, the authors seem to contradict their model with the statement, “Rather, they were related to fusion pores associated with the classical PSF features observed in amperometric monitoring. Therefore, it is not surprising that only small quantities were released,” Would this not mean the pore was closing? This paragraph should be more clearly written to show the speculation and the different angles to the argument.

---

## [Reviewer Report]

The authors have done a good job of answering the most recent questions. Only the discussion related to point 9 is one this reviewer disagrees with. The authors found a different reference (Nguyen et al., 2023) that I missed. The reference on nanoSIMS I referred to was Nguyen et al., 2022 (ACS Nano 2022, 16, 3, 4831–4842). In this manuscript they visualize partial release and, indeed, quantify it. They found closed vesicles with cytoplasmically located label in them following exocytosis and were able to quantify the 13C labelled dopamine in vesicles after stimulation showing it was approximately 60%. This is indeed counter to the statement that vesicles can stay open, but are missed as too little is released. It is worth a look.